

# Identification and validation of autophagy-related genes in Hirschsprung's disease

Ting Yao[1,*], Zenghui Hao[1,*], Wei Fan[1], Jinbao Han[1], Shuyu Wang[2], Zaiqun Jiang[1], Yunting Wang[1], Xiao Qian Yang[1] and Zhilin Xu[1]

[1] Department of Pediatric Surgery, The Six Affiliated Hospital of Harbin Medical University, Harbin Medical University, Harbin, Heilongjiang, China

[2] Department of Neurology, The First Affiliated Hospital of Harbin Medical University, Harbin Medical University, Harbin, Heilongjiang, China

[*] These authors contributed equally to this work.

## ABSTRACT

**Background**. Hirschsprung's disease (HSCR) is a congenital disorder characterized by aganglionosis in the intermuscular and submucosal nerve plexuses of the gut, leading to impaired gastrointestinal function. Although the precise cause and pathophysiology of HSCR remain elusive, increasing evidence points to a significant role of autophagy in its development, warranting further investigation into its underlying mechanisms.

**Methods**. This study utilized publicly available microarray expression profiling datasets, GSE96854 and GSE98502, from the Gene Expression Omnibus (GEO). The R software (version 4.2.0) was employed to identify autophagy-related genes potentially showing differential expression in HSCR. Subsequent analyses included correlation analysis, Gene Ontology (GO) and Kyoto Encyclopedia of Genes and Genomes (KEGG) pathway enrichment, and protein-protein interaction (PPI) network analysis using the STRING database (version 11.0) and Cytoscape software (version 3.8.2). Ultimately, HSCR samples were used to verify the mRNA levels of important genes by quantitative real-time polymerase chain reaction (qRT-PCR) in a laboratory setting.

**Results**. We have discovered 20 genes that are involved in autophagy and show variable expression. Among these genes, 15 are up-regulated and five are down-regulated. The enrichment analysis using the GO and KEGG pathways revealed a notable enrichment in pathways related to the control of autophagy. Nine hub genes were found *via* the investigation of the PPI network constructed from STRING database and module analysis using Cytoscape. Moreover, the concordance between SIRT1 expression in the HSCR model and the bioinformatics analysis of mRNA chip findings was validated using qRT-PCR.

**Conclusion**. Utilizing bioinformatics analysis, we identified 20 potential genes associated with Hirschsprung's disease that play a role in autophagy. Notably, the upregulation of SIRT1 may profoundly influence the progression of HSCR by regulating autophagy-related pathways, offering a novel perspective on the disease's pathogenesis.

Corresponding author
Zhilin Xu, xzlin333@163.com

## INTRODUCTION

Hirschsprung's disease (HSCR) is a congenital disorder characterized by the absence of nerve cells in the intermuscular and submucosal nerve networks of the intestines, leading to compromised gastrointestinal function (*Amiel & Lyonnet, 2001*). HSCR likely occurs due to the premature cessation of vagal neural crest cell migration from the head to the tail during weeks 5 to 12 of gestation. This interruption causes problems in the movement, growth, and specialization of intestinal neural crest cells (*Borrego et al., 2013*). Autophagy is a tightly regulated biological process that is critical for the degradation and recycling of cellular components, damaged organelles, and protein aggregates within cells (*Li et al., 2016*). Recent investigations have shown the crucial function of autophagy in preserving cellular stemness and promoting diverse stages of differentiation, such as cell proliferation and migration (*Foerster et al., 2021*). Autophagy is implicated in a variety of diseases, including intestinal problems. Nevertheless, the autophagy-related genes linked to HSCR have not been extensively studied and require more research.

In this study, we re-analyzed two public microarray datasets, GSE96854 and GSE98502, with a focus on autophagy-related genes. We used multiple bioinformatics approaches for integrated analysis, including functional enrichment and protein–protein interaction (PPI) network analyses. This provided insights into the underlying mechanisms of these genes. Finally, we identified nine hub genes (Fig. 1).

## EXPERIMENTAL PROCEDURES AND MATERIALS

### Autophagy-related genes datasets and microarray data

We acquired a cumulative of 232 genes linked to autophagy from The Human Autophagy Database (http://www.autophagy.lu/index.html). The mRNA expression profiles dataset from GSE96854 was acquired from the GEO database (http://www.ncbi.nlm.nih.gov/geo/). This dataset is part of the GPL18943 platform, specifically the NimbleGen Human Gene Expression 12x135K Array. It includes 30 colon samples, which were pooled into three control and three case specimens, each composed of tissues from 15 HSCR patients and 15 matched controls. The data were preprocessed using background correction and normalization through the RMA method. Quality control included inspecting MA plots and boxplots to ensure consistency across samples.

### Investigation of genes with altered expression associated with autophagy

The expression matrix was standardized, and the microarray data was obtained from the GSE96854 dataset. Next, the annotation procedure was carried out using the associated annotation files. The data in GSE96854 was verified for repeatability *via* the use of principal component analysis (PCA) using the R software package stats (version 4.2.0). Initially, we applied z-score normalization to the expression spectrum. Then, we conducted dimension reduction analysis using the prcomp function to generate a reduced matrix. The R software was used for differential expression analysis, employing the limma package (version 3.40.6) with the lmFit function for linear modeling. Adjusted *p*-values were calculated using
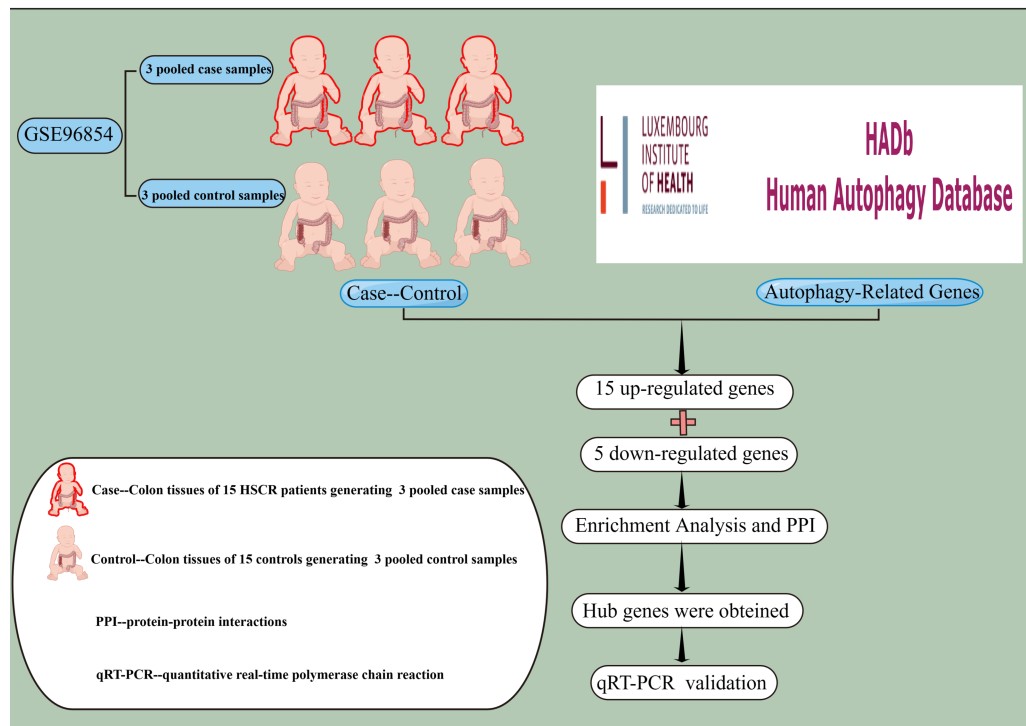

**Figure 1** **Experimental Design Roadmap.** The concept of experimental design was applied to extract gene expression profiles from colon tissues of 15 HSCR patients and 15 controls in the GSE96854 dataset. A total of 232 autophagy-related genes were collected from The Human Autophagy Database, out of which 15 up-regulated genes and five down-regulated genes were identified through differential analysis. Subsequently, enrichment analysis and PPI network construction led to the identification of nine hub genes. Finally, qRT-PCR was employed to validate the findings *in vitro* using colon tissues from HSCR patients and normal controls. PPI, protein-protein interaction; qRT-PCR, for quantitative real-time polymerase chain reaction. This image was drawn using Figdraw.

the Benjamini–Hochberg method. Statistical significance was assessed using an adjusted *P*-value threshold of less than 0.05 and an absolute fold-change value greater than 1.5. The R packages 'heatmap' and 'ggplot2' were used to create aesthetically pleasing heatmaps, volcano plots, and box plots.

## GO AND KEGG PATHWAY ENRICHMENT ANALYSIS OF GENES ASSOCIATED WITH AUTOPHAGY

In order to confirm the functional importance of prospective targets, a functional enrichment analysis was conducted. Gene Ontology (GO) is a commonly used technique for gene annotation, which includes the categorization of genes into cellular components (CC), molecular functions (MF), and biological processes (BP). The Kyoto Encyclopedia of Genes and Genomes (KEGG) is a database specifically created to facilitate the methodical examination of gene function and genomic information. It focuses on studying the interconnectedness of gene expression within a comprehensive network. GO and KEGG pathway enrichment analyses were performed using the clusterProfiler package (version

3.14.3) with a $p$-value cutoff of 0.05 and $q$-value cutoff of 0.05. For KEGG, organism-specific pathway annotations for *Homo sapiens* were used.

## Investigation of protein-protein interaction (PPI) and correlation analysis for genes associated with differential expression of autophagy

The differential expression analysis of autophagy-related genes was performed using the STRING database (https://string-db.org/) to construct a protein-protein interaction (PPI) network. The PPI network was visualized and analyzed using Cytoscape software (version 3.8.2). Key nodes within the network were identified based on their degree centrality (DC) scores, calculated with the cytoHubba plugin, which applies topological analysis to determine the most significant nodes and subnetworks. Our analysis identified nine hub genes with the highest centrality scores.. Spearman's correlation analysis of differentially expressed autophagy-related genes was performed using the 'corrplot' tool in the R program.

## Further validation of hub gene expression in HSCR colon specimens necessitates the inclusion of additional datasets

The mRNA expression levels of the hub genes that were discovered were confirmed in the GSE98502 dataset, which consists of eight control subjects and eight patients with HSCR. The datasets were compared using a $t$-test, and statistical significance was evaluated using a significance level of $P < 0.05$. RT-qPCR analysis was performed to assess the mRNA expression of central genes in both HSCR tissues and normal controls. This research received permission from the Ethics Committee of the Sixth Affiliated Hospital of Harbin Medical University. Written informed consent was obtained from the parents/guardians of all participants prior to inclusion in the study. This research investigated HSCR specimens obtained from 10 patients, consisting of eight males and two females. The patients' ages ranged from 1 to 14 months, with a median age of $5.9 \pm 3.9$ months. The specimens were collected from infants who had undergone transanal pull-through surgery. Additional details may be found in Table S1. Aganglionic specimens were collected from the colon of individuals with Hirschsprung's disease during the pull-through operation. The control samples consisted of normal colonic tissues obtained from individuals who had undergone anal atresia surgery and had a closed colostomy. The control group included nine male and one female patients, all aged between 6 and 12 months.

## RNA isolation and quantitative real-time PCR (qRT-PCR)

The extraction of RNA from the colon tissues was performed using Trizol (Invitrogen, Carisbad, CA, USA). The One Step SYBR® PrimeScript™ RT-PCR Kit II (Perfect Real Time; TaKaRa Biotechnology Dalian, China) was used to execute a one-step qRT-PCR. The experiment was conducted using a LightCycler 480 Real-Time PCR apparatus (Roche Diagnostics, Basel, Switzerland) following the instructions provided by the manufacturer. The qRT-PCR used the following primers (5–3):

*EGFR* (forward) CCCACTCATGCTCTACAACCC,
(reverse) TCGCACTTCTTACACTTGCGG;
*SIRT1* (forward) TGTGTCATAGGTT AGGTGGTGA,

(reverse) AGCCAATTCTTTTTGTGTTCGTG;
*CDKN2A* (forward) GGGTTTTCGTGGTTCACATCC,
(reverse) CTAGACGCTGGCTCCTCAGTA;
*ATG3* (forward) GACCCCGGTCCTCAAGGAA,
(reverse) TGTAGCCCATTGCC ATGTTGG.

## STATISTICAL ANALYSIS

The statistical analyses were performed using R software (version 4.2.0; *R Core Team, 2022*). The gene expression levels of our clinical samples were compared using a two-tailed Student's *t*-test, and a significance threshold of $P < 0.05$ was utilized to determine statistically significant differences.

## RESULTS

### Retrospective analysis of differential expression of autophagy-related genes in Hirschsprung's disease

We performed a comprehensive investigation of the expression levels of 232 autophagy-related genes across three pooled samples from HSCR patients and three pooled control samples. PCA was first conducted to assess the variability and repeatability of the data, ensuring robust clustering between HSCR and control samples (Fig. 2A). We identified 20 genes showing significant differential expression, with 15 up-regulated and five down-regulated, based on a significance threshold of $P < 0.05$ and an absolute fold-change greater than 2.0. These results were visualized using a heatmap to display overall gene expression patterns and a volcano plot to highlight the most significantly up- and down-regulated genes (Figs. 2B and 2C). Further analysis of these expression patterns was conducted through violin plots, which depict the distribution of expression levels for these genes across the HSCR and control groups (Figs. 3A and 3B). The five genes that showed increased expression were *NRG1, BAG3, CAPN2, ITGA6,* and *EGFR*. On the other hand, the five genes that showed decreased expression were *SERPINA1, BNIP3, NRG2, CDKN2A,* and *ITPR1*. These findings are shown in Figs. 2A and 2B, and summarized in Table 1.

### Correlation expression of differentially expressed autophagy-related genes

A detailed correlation analysis was conducted to explore the interactions between the expression levels of 20 differentially expressed autophagy-related genes in the GSE96854 dataset. The correlation coefficients were calculated using Spearman's method, with a threshold of 0.3 set to identify significant correlations. The analysis revealed complex interactions among these genes, with positive correlations identified between genes such as *NRG1, CAPN2, ITGA6, and EGFR*. In contrast, negative correlations were observed, such as between *NRG1* and *BNIP3*, as well as *NRG1* and *CDKN2A*. The first heatmap illustrates the overall correlation structure among the genes, while a second, more focused heatmap highlights genes with stronger correlations (Spearman's correlation coefficients >0.3). This heatmap (Figs. 4A and 4B) visualizes both positive and negative correlations using a color gradient, where red indicates positive correlations and blue indicates negative

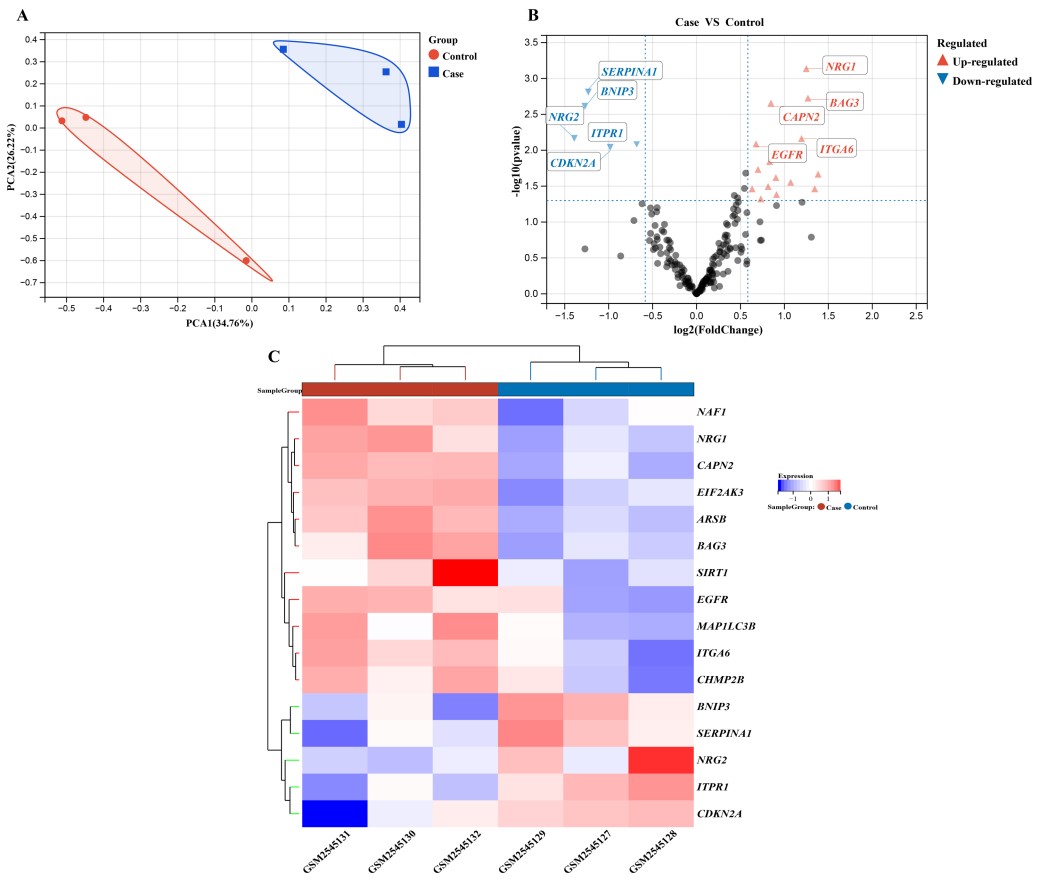

**Figure 2  Gene expression analysis visualizations: PCA, volcano plot, and hierarchical clustering heatmap.** Differentially expressed autophagy-related genes were identified in colon samples from cases and controls. (A) Principal component analysis was performed on GSE96854, with batch removal resulting in the intersection of two datasets that can be used for subsequent analyses as shown in the schematic diagram. (B) A volcano plot was constructed using fold change values >2.0 and $P < 0.05$. The red dots indicate genes that have been significantly up-regulated, while the blue dots represent those that have been significantly down-regulated. (C) Heatmap of 20 autophagy-related genes exhibiting differential expression between colon samples from cases and controls. The heatmap displays differential gene expression across various tissues, with different colors indicating distinct trends in gene expression.

correlations, with darker shades representing stronger correlations. This analysis is critical for identifying potential gene-gene interactions that may contribute to the regulation of autophagy in HSCR.

## Functional and pathway enrichment analysis of the differentially expressed genes associated with autophagy

In order to investigate the possible biological functions of DEGs, we used the R program to perform GO and KEGG enrichment analyses (Table S2). The GO analysis categorized DEGs into 919 biological processes (BPs), 65 molecular functions (MFs), and 112 cellular components (CCs), highlighting key processes such as autophagy, which utilizes macroautophagic mechanisms, and various cellular components including the endoplasmic

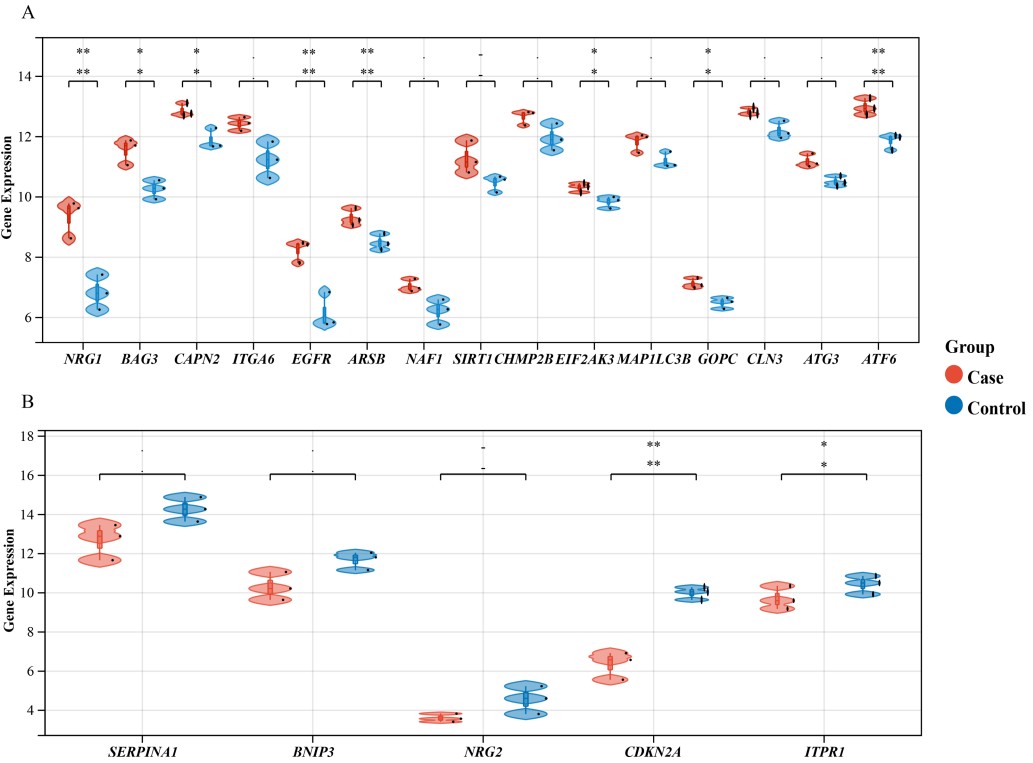

**Figure 3** **Differential Gene Expression Analysis: Case *vs.* Control Groups *via* Violin Plots.** Violin plots depict the expression levels of 20 autophagy-related genes that are differentially expressed in case and control samples. (A) The top 15 up-regulated autophagy-related genes show significant differential expression between case colon samples and control samples, as depicted by the violin plots. (B) The last five down-regulated autophagy-related genes also exhibit significant differential expression between case colon samples and control samples, as shown by the violin plots. Different colors represent different groups. Statistical significance is indicated by asterisks (*$P < 0.05$, **$P < 0.01$).

reticulum and organelle membranes. Additionally, molecular functions such as enzyme binding and cadherin binding were significantly enriched (Figs. 5A–5C). The KEGG pathway analysis further revealed significant enrichment in autophagy-related pathways, including cellular senescence and the FoxO signaling pathway (Figs. 6A and 6B), providing insight into the complex mechanisms underlying HSCR. These enrichments not only highlight the role of autophagy in HSCR but also suggest that pathways related to cellular senescence and FoxO signaling may contribute to the disease's pathogenesis, emphasizing the intricate interplay between different cellular processes.

## Exploration of protein-protein interaction networks and detection of central genes

To gain a deeper understanding of the interactions between differentially expressed autophagy-related genes, we constructed a PPI network using the STRING database. This network allowed us to visualize and analyze the complex interactions between the 20 identified DEGs, helping to identify key genes within the autophagy-related network. The nine hub genes with the highest degree centrality (DC) scores were identified using

**Table 1  The 20 differentially expressed autophagy-related genes in colon samples from cases and controls.**

| Gene Symbol | logFC | P-value | Adjusted P-value | T | Changes |
| --- | --- | --- | --- | --- | --- |
| NRG1 | 2.515266384 | 0.000342269 | 0.063319673 | 6.269035768 | Up |
| BAG3 | 1.290357777 | 0.002314947 | 0.107066292 | 4.555551479 | Up |
| CAPN2 | 1.248786297 | 0.000728997 | 0.067432254 | 5.548953225 | Up |
| ITGA6 | 1.19463117 | 0.006863156 | 0.251385681 | 3.717129017 | Up |
| EGFR | 1.0902745619 | 0.024006962 | 0.370107338 | 2.834354471 | Up |
| ARSB | 0.846961015 | 0.002203959 | 0.107066292 | 4.595465466 | Up |
| NAF1 | 0.832092075 | 0.014375713 | 0.29735965 | 3.187439413 | Up |
| SIRT1 | 0.814564303 | 0.032025328 | 0.390017682 | 2.639686981 | Up |
| CHMP2B | 0.69288678 | 0.038279749 | 0.393430755 | 2.52017479 | Up |
| EIF2AK3 | 0.67694849 | 0.008153049 | 0.251385681 | 3.591198144 | Up |
| MAP1LC3B | 0.63199153 | 0.034464467 | 0.390017682 | 2.590434178 | Up |
| GOPC | 0.559534035 | 0.02084943 | 0.356618735 | 2.930502318 | Up |
| CLN3 | 0.544895447 | 0.034039712 | 0.390017682 | 2.598747137 | Up |
| ATG3 | 0.467254333 | 0.046081265 | 0.426251697 | 2.396496742 | Up |
| ATF6 | 0.430002507 | 0.042619723 | 0.414981509 | 2.448505293 | Up |
| SERPINA1 | −1.594610937 | 0.014466145 | 0.29735965 | −3.183059631 | Down |
| BNIP3 | −1.36435527 | 0.010413534 | 0.275214823 | −3.415036548 | Down |
| NRG2 | −1.143834525 | 0.032292088 | 0.390017682 | −2.634114933 | Down |
| CDKN2A | −1.096129695 | 0.035839463 | 0.390017682 | −2.564230317 | Down |
| ITPR1 | −1.086387215 | 0.021204357 | 0.356618735 | −2.918958624 | Down |

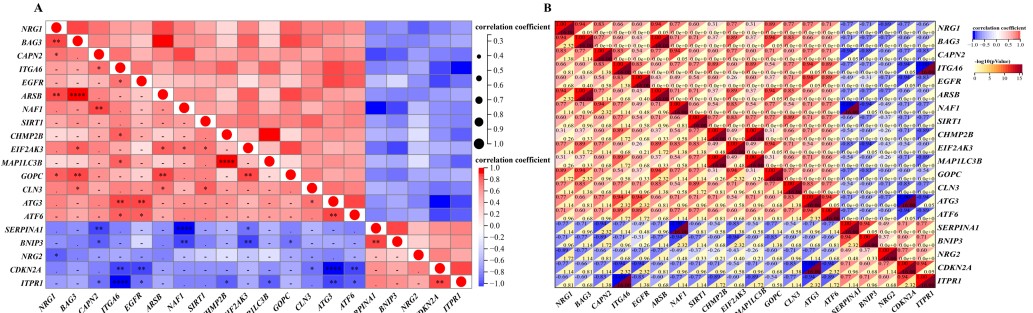

**Figure 4  Gene correlation matrices: visual and numerical analysis.** Spearman's correlation analysis of the 20 differentially expressed autophagy-related genes. (A, B), Correlation heatmap. The abscissa and ordinate represent genes, different colors represent different correlation coefficients (red represents positive correlation, and blue represents negative correlation). The darker the color, the stronger the relation. Asterisks (*) stand for significance levels, **** for $P < 0.0001$, ***$P < 0.001$, **$P < 0.01$, *$P < 0.05$.

Cytoscape (v3.8.2), indicating their potential importance in the network (Table 2). These hub genes include *EGFR, MAP1LC3B, SIRT1, EIF2AK3, ATF6,* and *ATG3,* which showed increased expression, and *ITPR1, CDKN2A,* and *BNIP3,* which showed decreased expression. The altered expression patterns of these hub genes may be closely linked to the

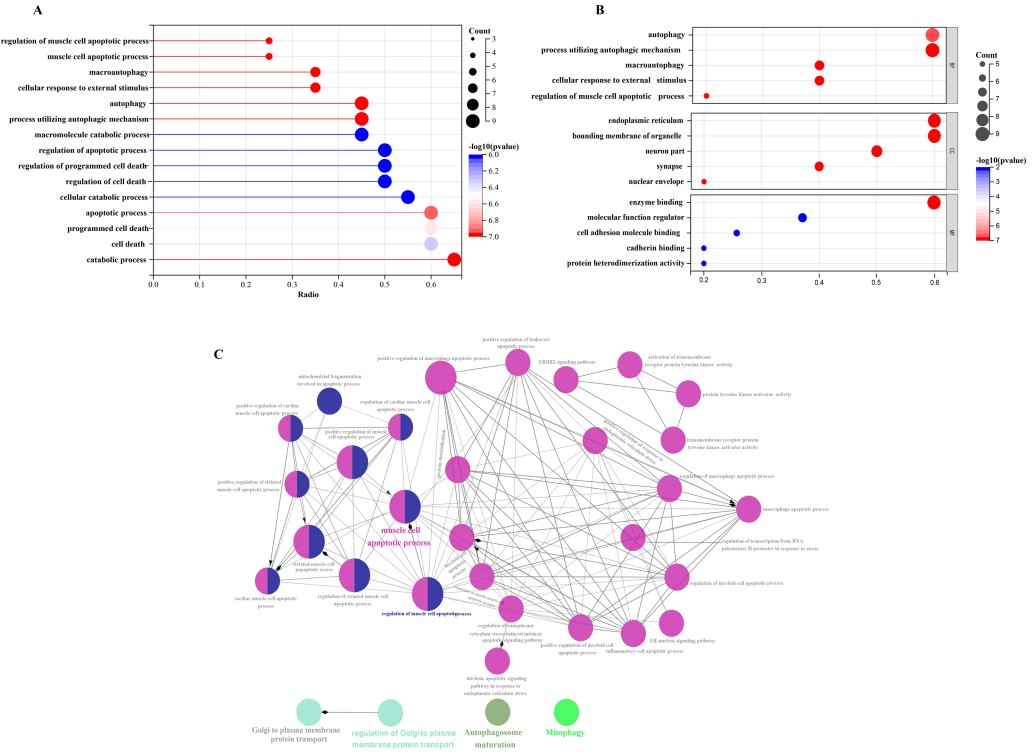

**Figure 5** **Comparative gene-function network analysis.** GO enrichment analysis of 20 differentially expressed autophagy-related genes, including BPs, CCs, and MFs. (A) Bar plot of enriched GO terms. (B) Bubble plot of enriched GO terms. (C) The visualization of 20 autophagy-related genes, and the functional group network constructed by the ClueGO plugin of Cytoscape. GO, Gene Ontology; BP, biological processes; CC, cellular components; MF, molecular functions.

development of HSCR, particularly through their roles in regulating autophagy (Figs. 7A and 7B).

## Verification of hub gene expression

We selected an HSCR dataset to validate the expression levels of these core genes. The findings revealed an increase in the expression of *EGFR, SIRT1, EIF2AK3,* and *ATG3* in contrast to normal intestinal tissue, although *MAP1LC3B* showed no change. Nevertheless, *ATF6* was suppressed. The bioinformatics study indicated earlier anticipated that these genes will be upregulated. The expression of *CDKN2A* was decreased, whilst the expressions of *ITPR1* and *BNIP3* were increased, in contrast to the anticipated outcomes from the bioinformatics analysis (Fig. 8). Out of the genes that were examined, only *EGFR, SIRT1, CDKN2A,* and *ATG3* showed statistical significance and concordance with earlier bioinformatics analysis ($P < 0.05$), which supports our previous results.

In order to confirm the trustworthiness of the GSE96854 dataset, we performed qRT-PCR analysis on four autophagy-related genes that showed varying levels of expression in clinical samples. The aganglionic colon tissue sample showed a significant increase in the expression levels of *EGFR, SIRT1,* and *ATG3* compared to the control sample. This finding

**A**

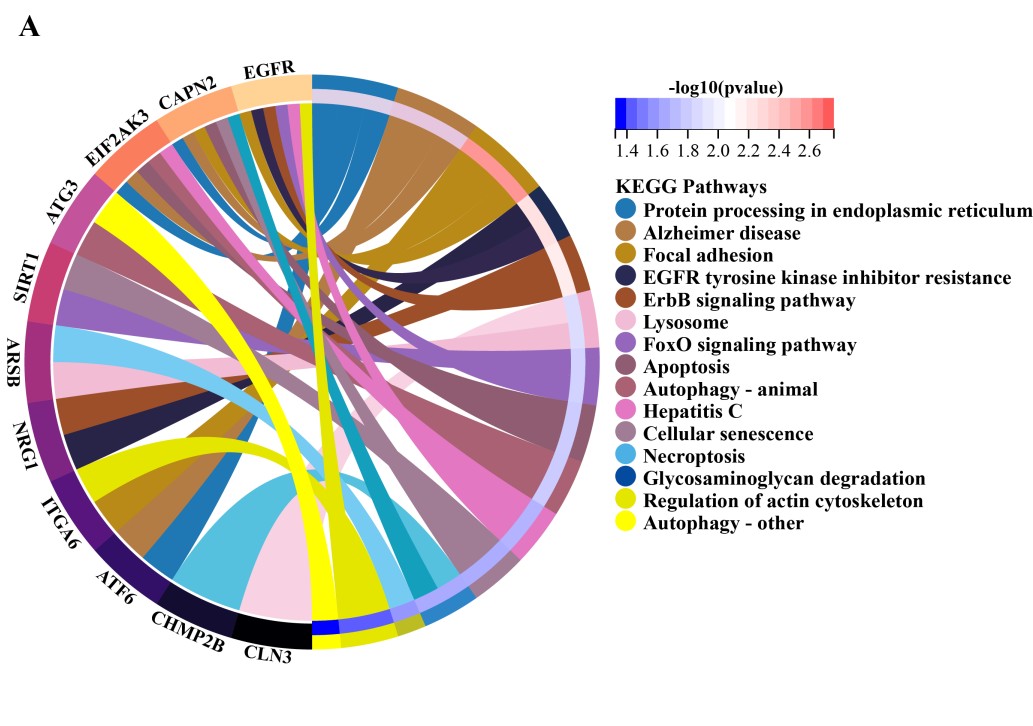

**B**

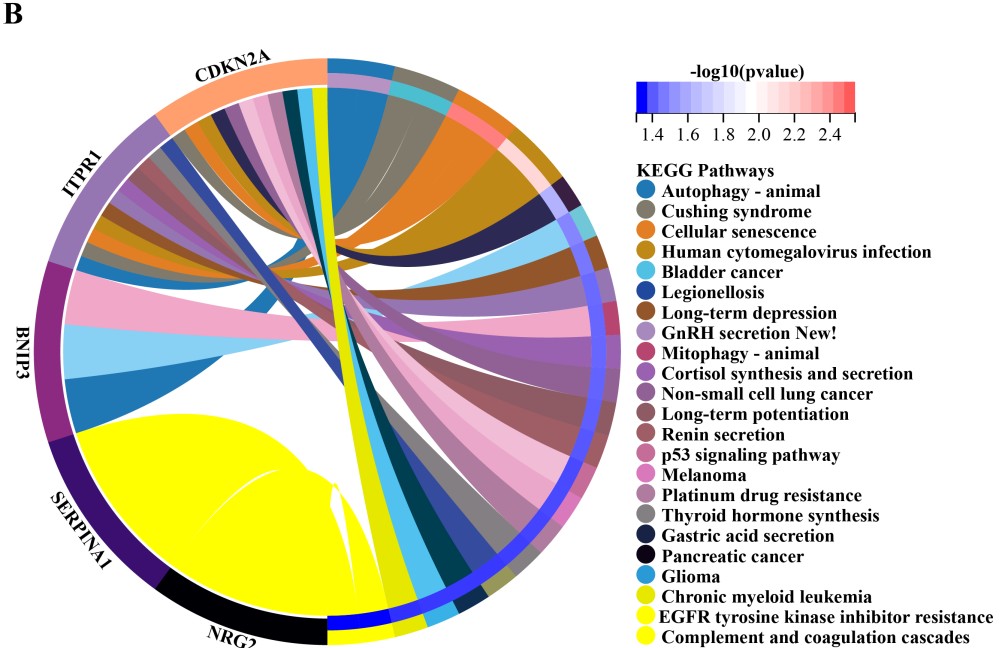

**Figure 6  Gene-pathway association analysis in circos plots highlighting significant biological processes.** KEGG enrichment analysis of 20 differentially expressed autophagy-related genes. (A) KEGG analysis of 15 up-regulated expressed autophagy-related genes. (B) KEGG analysis of five down-regulated expressed autophagy-related genes. KEGG, Kyoto Encyclopedia of Genes and Genomes.

**Table 2  Top nine in network ranked by DC method.**

| Rank | Gene ID | Gene name | Score | Changes |
|------|---------|-----------|-------|---------|
| 1 | EGFR | Epidermal growth factor receptor | 71.33333333 | Up |
| 2 | MAP1LC3B | Microtubule associated protein 1 light chain 3 beta | 63.26666667 | Up |
| 3 | SIRT1 | Sirtuin 1 | 20.13333333 | Up |
| 4 | EIF2AK3 | Eukaryotic translation initiation factor 2 alpha kinase 3 | 10.33333333 | Up |
| 5 | ITPR1 | Inositol 1,4,5-trisphosphate receptor type 1 | 6 | Down |
| 6 | ATF6 | Activating transcription factor 6 | 2.8 | Up |
| 7 | CDKN2A | Cyclin dependent kinase inhibitor 2A | 2 | Down |
| 8 | ATG3 | Autophagy related 3 | 1.066666667 | Up |
| 9 | BNIP3 | BCL2 interacting protein 3 | 1.066666667 | Down |

aligns with the findings obtained from mRNA microarray analysis of colon tissue samples. Moreover, there was a significant decrease in the levels of *CDKN1A* expression. However, the levels of *SIRT1* expression were the only ones that showed a significant difference between the two groups ($P < 0.01$) (Fig. 9).

## DISCUSSION

This study explored the pathogenic mechanisms of HSCR through comprehensive bioinformatics analysis, leading to the identification of nine hub genes and their potential roles in the disease. The significant upregulation of *SIRT1*, for example, indicates its pivotal role in modulating autophagy in HSCR. The correlation analysis suggests that *SIRT1* may influence key pathways such as cellular senescence and FoxO signaling, both of which are implicated in the regulation of autophagy. This highlights a potential feedback loop where *SIRT1* activation could promote autophagy, leading to cellular changes characteristic of HSCR.

Rather than focusing on general autophagy mechanisms, this study specifically identifies how autophagy-related genes, particularly *SIRT1*, contribute to HSCR pathogenesis. Previous studies have linked *SIRT1* to the regulation of autophagy in gastrointestinal diseases, where its role in maintaining cellular homeostasis is critical (*Fefelova et al., 2016*). Our findings support and extend this research by demonstrating that *SIRT1* upregulation in HSCR tissues may reflect an adaptive response to disrupted autophagy processes. The involvement of other hub genes such as *ATG3* and *EGFR* further suggests that these genes are integral to the dysregulation of autophagy in HSCR and could serve as potential therapeutic targets (*Wang et al., 2023*; *Zhang et al., 2017a*). For example, *Zhang et al. (2017a)* demonstrated that *SIRT1* is involved in regulating intestinal inflammation through autophagy modulation, providing a possible explanation for its upregulation in HSCR. In addition to *SIRT1*, other hub genes like *EGFR* have been implicated in autophagy-related pathways, particularly in cancer biology, where EGFR's role in autophagy regulation is well established (*Wang et al., 2023*). These results are consistent with studies showing that EGFR-mediated autophagy can promote cellular survival under stress conditions, which could explain its increased expression in HSCR tissues. Moreover, the unexpected

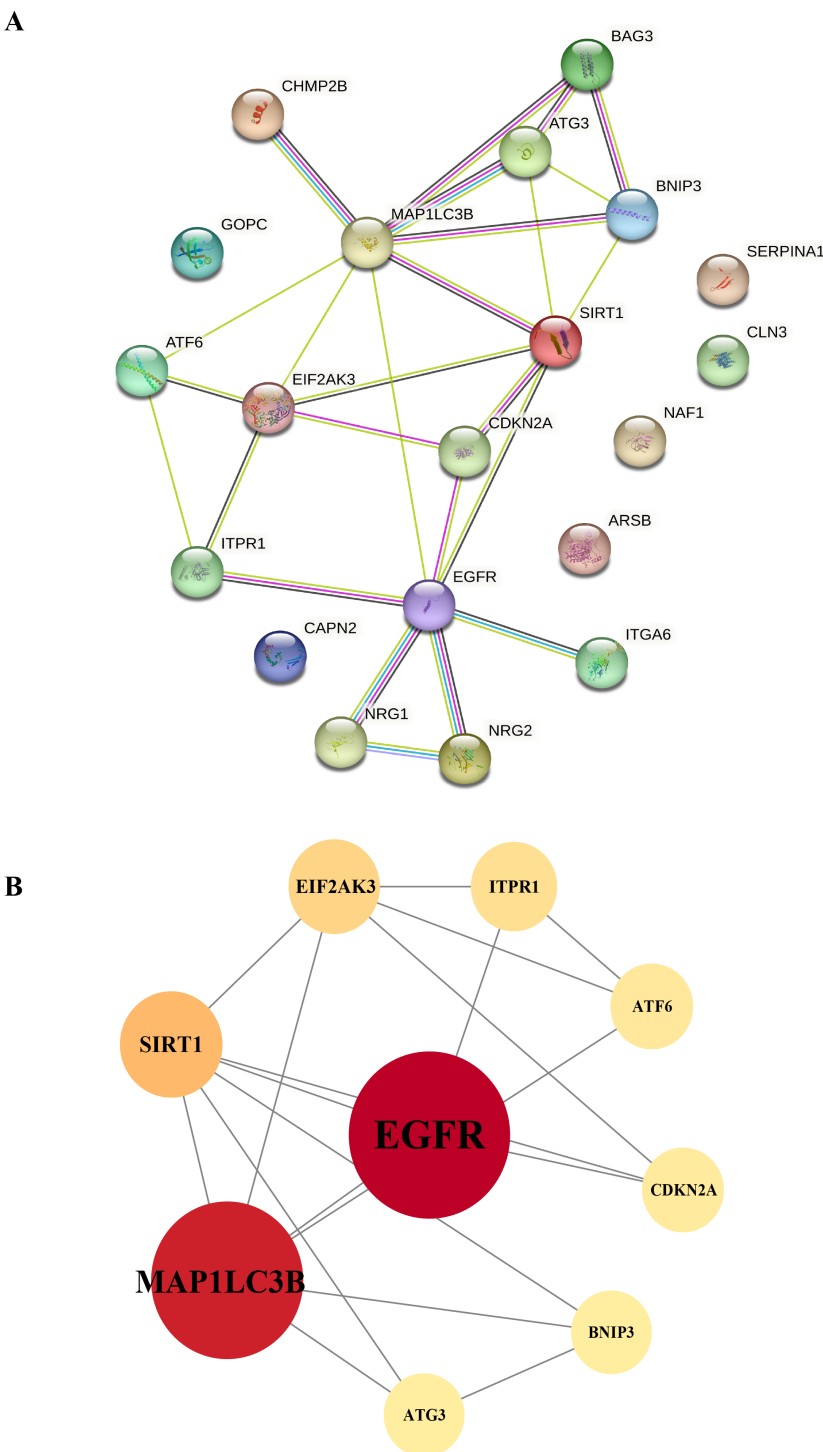

**A**

**B**

**Figure 7** **Protein-protein interaction networks.** Construction of the PPI network and identification of hub genes. (A) The PPI between 20 differentially expressed autophagy-related genes was constructed by using the STRING database. The node represents the gene, and the edge represents the relationship between the genes. (B) The top nine key genes were screened with the PPI network map. Different colors represent the size of the DC score for different genes; the deeper the color of the gene, the higher the score. PPI, protein–protein interaction.

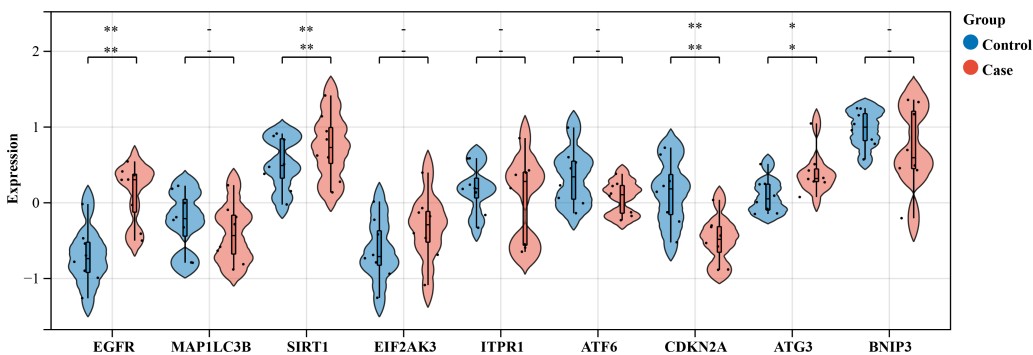

**Figure 8** **Violin plot analysis of gene expression in control *vs* case groups.** Validation of the nine hub genes' expression in external *HSCR* gene expression profile (GSE98502). HSCR, Hirschsprung's disease.

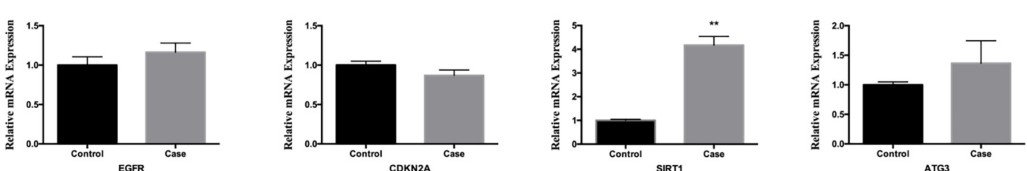

**Figure 9** **Comparative analysis of mrna expression levels for key genes in control and case study groups.** The mRNA level of four hub genes were measured in colon tissues. The mRNA level of *EGFR, CDKN2A, SIRT1*, and *ATG3* were evaluated in colon tissues as evaluated by qRT-PCR. *P*-values were calculated using a two-sided unpaired Student's *t*-test. *$P < 0.05$; **$P < 0.01$; unmarked, non-significant.

downregulation of *CDKN2A* despite predictions of upregulation based on bioinformatics models might be attributed to its role in a different cellular context, such as its known tumor-suppressive function in cancer (*Morigi, Perico & Benigni, 2018*). This discrepancy could reflect distinct regulatory mechanisms in the neural tissues affected by HSCR, where *CDKN2A* may play a less dominant role. These findings underscore the importance of conducting further experimental validation in specific tissue contexts to refine our understanding of these gene interactions.

Recent research has provided evidence that autophagosome structures are present in the myenteric plexus of individuals with HSCR who have narrow and transitional portions of the intestine. Furthermore, the expression of *Beclin1* and *LC3* was shown to be most prominent in the narrow segment (*Zhang et al., 2017b*). *Sbrana et al. (2016)* have shown the possible role of mir-939 in controlling *LRSAM1* in HSCR, and propose that autophagy may possibly play a part in the susceptibility to HSCR. Autophagy has been recognized as a crucial mechanism for maintaining normal gastrointestinal function in recent years. However, when autophagy becomes dysregulated, it may lead to gut dysfunction (*Li et al., 2020*). There is significant data suggesting that autophagy has a dual effect on gastrointestinal problems. Autophagy is a fundamental mechanism that occurs in all cells and helps maintain energy metabolism and facilitate substance recycling (*Qiang, 2016*). Nevertheless, its manifestation is swiftly increased in reaction to cellular energy deficiencies

resulting from insufficient dietary intake or circumstances such as hunger and hypoxia (*Guo et al., 2018*). Overstimulation of autophagy may cause excessive breakdown of vital cellular components, eventually resulting in autophagic cell death. *Zheng et al. (2021)* colleagues discovered that miR-222 was markedly increased in slow transit constipation (STC) mice compared to healthy control rats. Furthermore, they observed that miR-222 may stimulate excessive autophagy and death of isolated ICCs. Furthermore, it has been shown that the increase in miR-222 leads to heightened levels of autophagy-related proteins *LC3B* and *Beclin-1*. MiR-222's regulation of autophagy might result in tissue harm, presumably linked to gut dysfunction. *He et al. (2022)* found that electroacupuncture (EA) therapy may inhibit the PI3K/AKT/mTOR signaling pathway in the colonic tissues of mice with functional constipation (FC). This leads to the enhancement of autophagy in enteric glial cells (EGCs) and improvement in intestinal motility.

Autophagy is crucial in controlling the activity of different cell types in the intestinal milieu. This process involves both innate and adaptive immune cells, as well as stromal cells. Nevertheless, there is still a great deal of knowledge to be acquired about its precise role in preserving the integrity of the intestinal epithelial barrier. Autophagy is crucial in improving the death of intestinal cells and strengthening the function of the gut barrier. Autophagy has a crucial role in regulating cellular stress and preventing stress-induced apoptosis, which are important processes that cause the death of intestinal epithelial cells (IECs) (*Li et al., 2019*; *Wen et al., 2017*). Autophagy and autophagy-related proteins have a vital function in controlling intestinal damage and inflammation by regulating the process of epithelial cell apoptosis and necrosis (*Cong et al., 2017*; *Galluzzi et al., 2017*). Tumor necrosis factor inhibits autophagy, resulting in an increase in *CLDN2* expression. This, in turn, causes malfunction in epithelial tight junctions and leads to an increased permeability of the barrier (*Chen et al., 2021*). Furthermore, impairments in *Atg9, Atg1, Atg13*, and Atg17/FIP200 have been shown to cause a lack of autophagy in Drosophila, which subsequently leads to an elevated permeability of the intestinal barrier (*Morigi, Perico & Benigni, 2018*). Autophagy-related proteins have a crucial function in the host's response to infection in the intestinal epithelium. They do this *via* xenophagy, which involves the removal of foreign pathogens, as well as other processes that help regulate the immune system. This results in both the protection and vulnerability during intestinal infections (*Petri et al., 2017*). Multiple investigations have shown that the development of Hirschsprung-associated enterocolitis is caused by abnormalities in the structure and function of the intestinal mucosal barrier (*Petri et al., 2017*; *Ravanan, Srikumar & Talwar, 2013*). Ultimately, it is reasonable to hypothesize that autophagy may play a role in the development of HSCR. However, the investigation of autophagy-related genes using bioinformatics analysis has not been done in HSCR. Through bioinformatics research, we have discovered 20 putative autophagy-related genes in HSCR for the first time. GO and KEGG enrichment analysis were used to evaluate the probable biological roles of autophagy-related genes that were expressed differently. The findings of our study revealed that certain phrases were significantly linked to autophagy in animals, with a special emphasis on macroautophagy. In addition, by using PPI network and doing key module

analysis, we have discovered nine central genes that are linked to HSCR. These genes include *EGFR, MAP1LC3B, SIRT1, EIF2AK3, ATF6, ATG3, ITPR1, CDKN2A*, and *BNIP3*.

Out of the nine hub genes anticipated to be associated with diabetic retinopathy, only the expression of *SIRT1* was shown to align with the bioinformatics analysis of mRNA chip data. Prior research has shown the crucial function of the sirtuin protein family in controlling cellular metabolism and impacting many biological processes in several organs, such as the central nervous system, liver, pancreas, and intestinal tract (*He et al., 2022*). *SIRT1*, widely studied as a target of miR-132/-212, plays a vital role in the development of cancers, inflammatory diseases, neurological disorders, and other illnesses (*Doherty & Baehrecke, 2018*; *Li et al., 2020*). Although the exact cause-and-effect relationship between *SIRT1* and HSCR is not yet established, current research suggests a potential association between *SIRT1* and the development of HAEC. According to the findings of *Chen et al. (2021)*, exosomal miR-18a-5p has the ability to increase inflammatory responses and cause cell death in colonic epithelial cells in HAEC. This is achieved by activating the RORA-dependent SIRT1/NF $\kappa$B signaling pathway (*Chen et al., 2021*). Li et al. discovered that lipopolysaccharide (LPS) causes an increase in the expression of miR-132 and miR-212 in human aortic endothelial cells (HAEC), resulting in the inhibition of *SIRT1* and promoting the activation of the NLRP3 inflammasome (*Li et al., 2020*). The results emphasize the important function of the LPS/miR-132/-212/SIRT1/NLRP3 regulatory network in the development of HAEC. Our investigation revealed that *SIRT1* was increased in the narrow segments of HSCR, suggesting its involvement in the development of the condition.

Although we have made meticulous efforts, our research is limited by the inclusion of a very small number of clinical samples. Thus, it is crucial to verify our findings in a more extensive group of individuals diagnosed with HSCR. Another limitation of this study is the gender imbalance in the control group, which may introduce bias in the gene expression analysis. Future studies should aim for a more balanced sample to validate these findings. Moreover, while the expression levels of autophagy-related genes that are differently expressed have been validated in clinical specimens, additional research is required to understand the possible processes of these genes in HSCR cells and mice models.

## CONCLUSIONS

Through bioinformatics research, a total of 20 candidate genes associated with autophagy were discovered in HSCR. Nine hub genes, including *EGFR, MAP1LC3B, SIRT1, EIF2AK3, ATF6, ATG3, ITPR1, CDKN2A*, and *BNIP3*, were found by creating a PPI network and identifying important modules. This study found an increase in the expression of *SIRT1* in HSCR tissues and *in vitro* experiments confirmed that it can regulate autophagy, suggesting its potential role in HSCR pathogenesis by regulating autophagy-related pathways. Additional trials are required to investigate the regulatory role of *SIRT1* in the development of HSCR.

### Funding

This work was supported by the Construction Project for Pediatric Teacher Funding Project of Harbin Medical University (Grant number 0201-31021220022) and the Basic Research Business Fee Research Project of Provincial Higher Education Institutions in Heilongjiang Province in 2022- Medical Clinical Youth Science Research Project. The funders had no role in study design, data collection and analysis, decision to publish, or preparation of the manuscript.

### Grant Disclosures

The following grant information was disclosed by the authors:
The Construction Project for Pediatric Teacher Funding Project of Harbin Medical University: 0201-31021220022.
Basic Research Business Fee Research Project of Provincial Higher Education Institutions in Heilongjiang Province in 2022- Medical Clinical Youth Science Research Project.

### Competing Interests

The authors declare there are no competing interests.

### Author Contributions

- Ting Yao conceived and designed the experiments, performed the experiments, analyzed the data, prepared figures and/or tables, authored or reviewed drafts of the article, and approved the final draft.
- Zenghui Hao conceived and designed the experiments, performed the experiments, analyzed the data, authored or reviewed drafts of the article, and approved the final draft.
- Wei Fan performed the experiments, analyzed the data, authored or reviewed drafts of the article, and approved the final draft.
- Jinbao Han analyzed the data, prepared figures and/or tables, and approved the final draft.
- Shuyu Wang analyzed the data, prepared figures and/or tables, authored or reviewed drafts of the article, and approved the final draft.
- Zaiqun Jiang analyzed the data, prepared figures and/or tables, and approved the final draft.
- Yunting Wang conceived and designed the experiments, authored or reviewed drafts of the article, and approved the final draft.
- Xiao Qian Yang analyzed the data, prepared figures and/or tables, and approved the final draft.
- Zhilin Xu conceived and designed the experiments, performed the experiments, prepared figures and/or tables, authored or reviewed drafts of the article, and approved the final draft.

## Human Ethics

The following information was supplied relating to ethical approvals (*i.e.*, approving body and any reference numbers):

Ethics Committee of the Sixth Affiliated Hospital of Harbin Medical University

## Data Availability

The clinical data of patients with HSCR are available in the Supplementary Files.

## Supplemental Information

Supplemental information for this article can be found online at http://dx.doi.org/10.7717/peerj.18376#supplemental-information.

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
