# Peer review of "Identification and validation of autophagy-related genes in Hirschsprung’s disease"

_PeerJ, doi:10.7717/peerj.18376_

## Round 0.1 · original submission · Major Revisions

Based on the comprehensive reviews received, a decision has been made to require major revisions to the manuscript before it can be considered for publication. While the manuscript presents valuable findings, particularly regarding the role of autophagy in Hirschsprung's disease (HSCR), significant improvements are needed to enhance clarity, reproducibility, and overall scientific rigor.

**Reviewer 1** recommended acceptance, noting that the findings are interesting and should be reported in the literature. However, no specific issues were raised, so no immediate action is required based on this review.

**Reviewer 2** also recommended acceptance, appreciating the manuscript's efforts in identifying genes related to HSCR. The reviewer suggested ensuring consistency in the use of initials and abbreviations throughout the text, which the authors should address.

**Reviewer 3** highlighted several major issues requiring revision, particularly concerning basic reporting, experimental design, and the validity of the findings. The reviewer pointed out that the figures in the manuscript are of poor resolution with inconsistent font sizes, which impedes readability. Additionally, the reviewer emphasized the need to provide the code used for bioinformatics analysis to ensure reproducibility. The language used in certain sections of the manuscript requires refinement for clarity, and there are formatting errors, such as sentences ending with double periods, that need correction. Furthermore, the introduction should provide a more detailed justification for the study, and the discussion should be streamlined to focus on the study's key findings while integrating them with existing literature. The authors must address these significant issues to improve the manuscript's clarity, rigor, and impact.

**Reviewer 4** suggested minor revisions, particularly regarding the readability of Figure 5c and minor language improvements. The reviewer also raised concerns about potential bias in the control group and the possibility of comorbidities or genetic abnormalities in patients with anal atresia, which should be discussed as a potential limitation in the study.

To proceed, the authors should take the following actions: enhance the quality and consistency of all figures, including refining Figure 5c for better readability; provide the necessary code to ensure the reproducibility of the bioinformatics analysis; improve the clarity and language in the manuscript, addressing any formatting issues; offer detailed methodological explanations, including specific steps in data preprocessing and parameters used for analyses; and ensure that the discussion section is focused and well-integrated with existing literature, while acknowledging any potential biases or limitations in the study. Additionally, the introduction should be expanded to include a more detailed justification for the research, emphasizing its implications for clinical practice.

These revisions are essential for the manuscript to meet the journal's standards and to fully realize its potential contribution to the understanding of HSCR. We look forward to receiving a revised version that comprehensively addresses these points.

·

Basic reporting

valid

Experimental design

valid

Validity of the findings

valid

Additional comments

none

·

Basic reporting

Passed

Experimental design

Passed

Validity of the findings

Passed

Additional comments

Please check the use of initials and abbreviation

·

Basic reporting

Overall, the manuscript is written in a technically appropriate manner. However, it would benefit from some revisions to improve clarity and conciseness, especially in complex sections.
Major issues with basic reporting are as follows:
1. The figures in the main manuscript suffer from poor resolution and inconsistent font sizes, which significantly impede readability. The low-quality resolution blurs critical details, while the varying font sizes make it difficult to interpret labels and annotations accurately. This hinders the overall clarity and comprehension of the presented data. For example: Figure 4.
2. Code used to generate figures and execute the bioinformatics workflow for quantifying results from the datasets and downstream analyses has not been provided, thereby impeding the reproducibility of the findings.

Minor issues with basic reporting are as follows:
1. Certain sections could benefit from improved clarity and precision in language. For instance, sentences on lines 13-14, 17-18, 20-21, 24-25, 27-28, 31-32, 34-35, 41-42, 49-50, 53-54, 148-150, 165-167, 173-176, 219-222 have phrasing that could be revised to enhance comprehension.
2. The manuscript has some basic formatting issues that need to be addressed. For instance, there are sentences that end with two periods instead of one. Correcting these small errors will help improve the overall professionalism of the document.
3. A more detailed justification for the study could strengthen the introduction. Specifically, expanding on the potential implications of the findings for clinical practice would provide a more compelling rationale for the research.
4. The results section contains multiple visual representations of the same data (e.g., heatmap, volcano plot, and violin plots) without explaining the unique contribution of each figure. Clarify the unique insights each figure provides to avoid redundancy and enhance understanding.

Experimental design

Major issues with experimental design are as follows:
1. The manuscript specifies the R packages used for each of the analyses but does not include the exact functions or parameters for a function or the versions of the packages employed in some cases. This omission impedes reproducibility, as different versions of packages may yield varying results, and the absence of function details makes it challenging to replicate each analysis precisely.
2. Lines 60-63: The description of how the mRNA expression profiles dataset from GSE96854 was acquired and processed is too brief. It should include more details about the specific steps taken to ensure data quality and normalization procedures. I would suggest providing a step-by-step account of how the data was preprocessed, including any quality control measures, normalization techniques, and the rationale behind choosing specific methods.
3. Lines 78-85: The explanation of Gene Ontology (GO) and Kyoto Encyclopedia of Genes and Genomes (KEGG) pathway enrichment analysis is too general and lacks specific details on how these analyses were conducted. I would suggest providing detailed information on the tools and parameters used for GO and KEGG enrichment analyses, including any specific settings or thresholds applied during the analyses.
4. Lines 88-95: The process of constructing the protein-protein interaction (PPI) network and selecting key nodes based on Betweenness (DC) scores is mentioned but not thoroughly explained. I would suggest elaborating on any param the PPI network was constructed, including the specific criteria for selecting interactions, and provide a clear explanation of how the DC scores were calculated and used to identify key nodes.

Minor issues with experimental design are as follows:
1. While the statistical methods are well-described, providing more detailed information about the adjustment for multiple testing (e.g., Bonferroni correction) and the rationale behind the choice of specific statistical tests would further strengthen the methodology section.
2. Lines 129-131: The manuscript mentions using the Student's t-test with a significance threshold of P<0.05 but does not specify whether this is a one-tailed or two-tailed test. Specify the type of t-test used to enhance the clarity of the statistical analysis.

Validity of the findings

Major issues with the results and discussion section are as follows:
1. Lines 151-160: The correlation analysis results are mentioned without sufficient clarity on the interpretation of the correlations and the criteria for selecting significant correlations. I would suggest providing more detailed information on how the correlation analysis was conducted, the significance thresholds used, and a clearer interpretation of the correlation heatmap.
2. Lines 204-210: The discussion section has redundant information about autophagy and its general roles without focusing specifically on how the findings of this study contribute to the existing knowledge about HSCR. I would suggest streamlining the discussion to focus on the key findings of this study, how they advance the understanding of HSCR, and avoid redundant general information about autophagy.
3. Lines 211-220: The discussion does not sufficiently integrate the study’s findings with existing literature to provide a comprehensive understanding of the results. I would suggest discussing the results in the context of existing studies, highlighting similarities, differences, and novel insights provided by this research.
4. Lines 223-229: The manuscript mentions findings that differ from bioinformatics predictions (e.g., expression levels of certain genes) without providing possible explanations or implications. I would recommend offering potential explanations for any unexpected findings and discussing their implications. Explore whether these differences could be due to methodological differences, sample variability, or biological reasons.

Minor issues with the results and discussion section are as follows:
1. Lines 151-160: There is inconsistency in terminology, such as switching between "correlation analysis" and "correlation heatmap." Use consistent terminology throughout to avoid confusion.
2. Lines 204-210: The discussion includes broad statements about the role of autophagy in various diseases without linking these statements specifically to the study's findings. Focus the discussion on how the findings of this study specifically contribute to the understanding of HSCR and autophagy.
3. Lines 223-229: The manuscript mentions the biological implications of autophagy-related genes without providing a detailed explanation. I would suggest elaborating on the biological implications of the findings, explaining how they contribute to the understanding of HSCR pathogenesis.

Additional comments

The manuscript presents some valuable insights into the role of autophagy in HSCR, but all the issues outlined above must be addressed to enhance the clarity, rigor, and impact of the study. I recommend a thorough revision to address these concerns before the manuscript can be considered for publication.

·

Basic reporting

minor English revision is needed
raw data is shared and displayed adequately with graphs and various plots
figure 5c needs to be refined since the labels cannot be read

Experimental design

method is designed meticulously, although
some points to point out
1. the control group consisted of males primarily (line 112), could this be a potential bias?
2. the control group consisted of patients who suffered from anal atresia, although i couldnt find any references in the bibliography contradicting the correlation with hirschsprung disease with anal atresia, there could be a possibility of comorbidities or genetical abnormalities in that control group. it should be pointed out as a possible limitation of the study

Validity of the findings

no comment

---

## Round 0.2 · accepted · Accept

The manuscript has been substantially improved following the revisions. The major and minor issues raised in the initial review have been addressed, resulting in a clearer, more focused, and methodologically sound manuscript. The study offers valuable insights into the role of autophagy in HSCR and is now recommended for publication.

·

Basic reporting

The manuscript is now considerably improved in terms of clarity and professionalism. The authors have made significant revisions in response to the major and minor issues highlighted in the initial review.
1. Figures: The authors have improved the resolution and consistency of the figures. The enhancements to Figure 4, in particular, are notable, making the data more interpretable and clear. This addresses the previous concerns about poor resolution and inconsistent font sizes.
2. Code Availability: The authors have provided the necessary code as a supplement, ensuring reproducibility of the bioinformatics analyses. This is a significant improvement that was crucial for validating the study’s findings.
3. Language Clarity: The manuscript has undergone substantial revisions for clarity and conciseness, particularly in complex sections. Phrasing issues highlighted in specific lines have been addressed, making the text more comprehensible.
4. Justification for the Study: The introduction now includes a more detailed justification for the study, particularly in terms of its implications for clinical practice. This provides a stronger rationale for the research.
5. Results Section: The redundancy in the visual representations of data has been clarified. The unique contribution of each figure has been explained, reducing redundancy and enhancing the reader’s understanding.

Experimental design

The experimental design has been refined with additional details, addressing the major concerns from the initial review.
1. R Packages and Parameters: The authors have included details about the specific functions, parameters, and versions of R packages used, which were missing previously. This improvement enhances the reproducibility of the analyses.
2. Data Preprocessing: The description of how the mRNA expression profiles were processed has been expanded. The authors provided some details on the data preprocessing, including quality control and normalization procedures.
3. GO and KEGG Pathway Analysis: The manuscript now includes more detailed information about the tools and parameters used for Gene Ontology (GO) and KEGG pathway enrichment analyses. This additional detail is crucial for understanding how these analyses were conducted.
4. PPI Network Construction: The process of constructing the protein-protein interaction (PPI) network and selecting key nodes based on Betweenness scores has been explained, addressing the previous gaps in the manuscript.
5. Statistical Methods: The statistical methods, including the adjustment for multiple testing and the rationale behind specific tests, have been clarified. This strengthens the methodological rigor of the study.
6. Student's t-test: The manuscript now specifies that a two-tailed Student's t-test was used, addressing the ambiguity noted in the initial review.

Validity of the findings

The revisions made in the results and discussion sections have enhanced the validity and impact of the findings.
1. Correlation Analysis: The correlation analysis results have been clarified, with detailed information on the interpretation of correlations and the criteria for significance. This provides a clearer understanding of the findings.
2. Discussion Section: The discussion has been streamlined to focus more on the key findings and their contributions to the existing knowledge about HSCR. Redundant information about autophagy has been removed, making the discussion more focused and relevant.
3. Literature Integration: The discussion now better integrates the study’s findings with existing literature, providing a more comprehensive understanding of the results.
4. Unexpected Findings: The manuscript now discusses potential explanations for unexpected findings, such as the differences between bioinformatics predictions and observed gene expression levels. This consideration of alternative explanations adds depth to the analysis.
5. Terminology Consistency: The terminology in the manuscript, particularly in the correlation analysis, has been made consistent, reducing potential confusion.
6. Discussion of Biological Implications: The manuscript now elaborates on the biological implications of the findings, particularly how they contribute to the understanding of HSCR pathogenesis. This adds value to the discussion.

Additional comments

The manuscript presents valuable insights into the role of autophagy in HSCR. The major and minor issues outlined in the initial review have been addressed, resulting in a manuscript that is clearer. I recommend this revised manuscript for publication.